# The Foundation of Classical Mechanics

**Danilo Capecchi** 

Department of Structural and Geotechnical Engineering, Sapienza University of Rome, 00185 Roma, Italy; danilo.capecchi@uniroma1.it

**Definition:** Mechanics is the science of the equilibrium and motion of bodies subject to forces. The adjective classical, hence Classical Mechanics, was added in the 20th century to distinguish it from relativistic mechanics which studies motion with speed close to light speed and quantum mechanics which studies motion at a subatomic level.

**Keywords:** classical mechanics; fundaments; history; epistemology; analytical mechanics

## 1. Introduction

*Classical mechanics*, or more commonly *Mechanics*, is the discipline devoted to the study of the equilibrium and motion of bodies subject to forces; the adjective *classical* sets it apart from *Relativistic mechanics* which studies motion with speed close to light speed and *Quantum mechanics* which studies motion at a subatomic level. From a historical perspective *Classical mechanics* is the form of mechanics after Newton; for this reason it is often referred to, though quite improperly, as *Newtonian mechanics*. For the 16th and 17th century one speaks of *Early classical mechanics* while for previous periods the locution *Ancient mechanics* is often used.

What is now called mechanics has played a preponderant role in the development of science since ancient Greece [1]. To explain its influence it is not enough, as done just above, to define mechanics as the reasoned study, that is a science, of the phenomena of equilibrium and motion. First of all, mechanics knows how to measure the phenomena of motion: in other words, however complex their appearances may be, whatever qualitative aspects they reveal, whether it is the changing figure of a cloud, of a waterfall, deformations and resistances of an elastic solid, mechanics knows how to define entirely with the help of numbers these motions, these resistances, these deformations. It is a quantitative discipline.

As for the influence of mechanics on the development of other sciences, it is not difficult to see the reasons for this. First of all, the phenomena of motion or of equilibrium occur constantly and everywhere, whether they appear alone or whether they are accompanied by other more complex phenomena (electrical, chemical, etc.). Mechanics was therefore the necessary basis for other sciences, at least as soon as they wanted to be sufficiently precise.

There was not a historical accident which gave mechanics its preponderance. Among all the phenomena, those which are the least difficult to measure, that is to define completely with numbers, are the phenomena of motion: it is therefore mechanics which became the first of all sciences to take a quantitative form. However, by virtue of the more abstract and geometric nature of the phenomena it studies, it had to vegetate as long as it was not in a condition to assume a quantitative form in all its aspects. Let us follow, for instance, a projectile launched into the air; what will we be able to say precisely about its motion if there is no way to register it? It is therefore understandable that the development of mechanics, or better of that part which studies motion, was at the same time so late and so prodigiously rapid once it had begun, and it immediately took precedence over other sciences, even over those which, like alchemy in the Middle Ages, seemed to precede it ([2], pp. 1–4).

Classical mechanics is today generally considered a mathematical physics discipline and as such of little interest to physics researchers. It is a mathematical theory whose

axioms, although they draw inspiration from the physical world, are considered as given; the only developments that can be expected are discoveries of new theorems more or less interesting for the applications of the discipline. It must be said that this point of view is not unassailable and the historical analysis of the discipline provides many elements to counter it.

> Despite the unusually distinguished and successful role Newtonian [classical] mechanics has played in the history of modern science, its foundations have been under vigorous debate since Newton first formulated his laws of motion. Moreover, although the axioms have received much more than two centuries of critical attention from outstanding physicists and philosophers, there still is wide disagreement about what they assert and what their logical status is. The axioms (or their logical equivalents) have been claimed to be either a priori truths, which can be asserted with apodictic certainty; [or] to be necessary presuppositions of experimental science though incapable either of demonstration by logic or refutation by observation; or to be empirical generalizations, "collected by induction from phenomena" ([3], p. 174).

In the 18th century, a century in which classical mechanics had reached a mature status, many important scientists like Euler and d'Alembert believed that mechanics were an *a priori* science, that needed no recourse to experience to be established. Otherwise other scientists, such as Daniel Bernoulli for instance, supported the idea of mechanics as an empiric discipline.

The possibility of this long dispute depended on the essence of mechanics. Even those who declared mechanics as an experimental science referred to the experience of everyday life and not to complicated experiments carried out in the laboratory, as was the case, even in the 18th century, for electricity and magnetism or thermology. For instance the law of lever was based on the observation that the more the heavier a body, the more it acts on an arm; the law of motion was based on the constant effect of a cause and that heavy bodies felt downward. The principle of inertia was stated based on simple thought experiments and the principle of action and reaction seemed to be self-evident. Of course no one denied the role of experience when a mechanical theory was be applied. For instance, to evaluate the law of falling of a heavy body, it is not enough to say that its speed varies proportionally to time. It is also necessary to know the value of the constant of proportionality, which is the acceleration of gravity. This can be known only by means of accurate experimentations in a laboratory. Equally the evaluation of masses, forces, times, distances may require sophisticated experiments and instruments.

Up to now no shared formulations of a completely axiomatized formulation of classical mechanics have been provided, even though some attempts have been made [4–7]. Commonly incomplete axiomatized versions circulate in which in some cases the difference between axioms and theorems is not completely clear. For example, Newton's second law of motion, $f = ma$, can be treated as a principle, a relationship between force $f$ and acceleration $a$, in which both members of the relationship are considered as primitive terms of the theory, or it can be treated as a simple definition of force. This matter of fact strongly suggests that mechanics needs some more studies by physicists in the attempt to improve its comprehension.

The difference between the two kinds of axiomatic formulations, the complete and the incomplete one, is not only of a quantitative but also epistemological nature. A complete axiomatic theory is by definition a closed system, straightforwardly formalizable with the sufficiently sophisticated language of mathematical logic so that it can contain most of mathematics. It has by definition a hypothetical deductive nature; that is, no truth value is given to the axioms of the theory. What matters is that the theory is coherent, even if this coherence as argued by Gödel's theorems cannot be proved but only assumed in the absence of evident inconsistencies. Primitive terms, definitions, axioms, have no definite meaning in themselves, even if they are given names used in everyday life, such as force, mass, inertial system. These quantities take on values through some rules of correspondence, connecting

the real world with the theory, which include all the conceptual difficulties of the physical theory that are avoided in the formalized theory. If the theory and the correspondence rules provide results corresponding to the empirical measurements, the theory is said to be validated (see the next sections).

In incomplete axiomatized theories, one starts from some axioms, generally treated as empirical laws and therefore considered true as such. From these axioms one derives conclusions that are true in themselves because they are deduced by true axioms with the indubitable laws of logic. If it were to be verified that the conclusions were not empirically true, the blame would not be attributed to the theory but to those who applied it, not having been able to take into account the precise manner of the different parameters involved. The incomplete axiomatic approach of contemporary treatments is from a logical point of view perfectly equivalent to the approach of Hellenistic Greece scholars, of Archimedes in particular, as will be clear from the next section. From certain points of view, therefore, it can be said that the essence of mechanics has remained unchanged over a period of more than two thousand years.

There are various formulations of classical mechanics. Perhaps the most important distinction is between a vector formulation that emphasizes the notion of force and refers to Newton and Euler and an analytic formulation that emphasizes the related notions of work and energy and refers to Leibniz and Lagrange. Up to now the equivalence of the two formulations has not been proven in a shared way, even if most scholars are convinced of it, or if they are not convinced they do not consider the fact particularly interesting. It must be said that many textbooks, even of a high level, try to prove this equivalence, unfortunately with some logical leaps. The few supporters of the non-equivalence between analytic and vector theories point out that within the analytic formulation, work and energy are concepts that can be easily derived from thermodynamics, which is not true for the vector formulation.

## 2. Ancient Mechanics

In ancient Greece mechanics was the name of the science dealing primarily with the study of equipments or machines (in Greek $\mu\eta\kappa\alpha\nu\eta$), to transport and lift weights. The search for equilibrium was not of practical interest and mechanics, at least at the beginning, did not care for it. From this point of view mechanics was different from modern statics which is instead seen as the science of equilibrium.

Mechanics with astronomy, music and optics was considered as a mixed mathematical discipline (Renaissance nomenclature), because it dealt with both mathematics and physics. Astronomy which studied phenomena of which only a limited direct experience could be shown, was a hypothetical deductive theory (modern terminology). That is a theory based on a principle assumed as hypothetical—for instance the diurnal rotation of the fixed stars sphere—that is not true for sure. The theory, but not the principle, was validated by its adherence to empirical observations. Different theories based on a different principle could save the same phenomenon. Mechanics, with music and optics, was instead based on indubitable axioms derived either from natural philosophy, or metaphysics or experience.

Certainly there have been very ancient reflections about mechanics; hardly any trace remains of them, however. An ancient mathematician credited with some theorizations was Archytas of Tarentum (fl. 400 BC) but the first known writing in the West is the *Mechanica problemata* attributed to Aristotle [8,9]. The attribution is doubtful, sometimes accepted, sometimes rejected by historians. Certainly a modern reader is surprised to see how a philosopher not particularly interested in mathematics, Aristotle, had taken an interest in mechanics.

The style of the *Mechanica problemata* is not the sober one of mathematicians and the fact that some scholars have seen in it the demonstration of the law of the lever while others had not, testifies to the ambiguity of the text. The first proofs of the law of the lever truly accepted by ancient mathematicians were those of Euclid and Archimedes. That of the

latter is still considered by many scholars as the most convincing even if some controversial points remain, as happens in any proof of the 'axioms' of a science.

Euclid is credited as the author of a text known in the West as *The book of the balance* ([10], pp. 24–30); Archimedes was the author of the *De planorum aequlibriis* [11]. Both the authors developed a theory of a deductive kind based on some simple (empirical?) axioms. Euclid, or better the pseudo-Euclid, assumed:

1.  When two equal weights are on the end of a lever of equal arms, the lever remains in equilibrium.
2.  If two weights keep a lever in equilibrium, the lever remains in equilibrium when the weights are moved along vertical wires ([10], p. 24).

Archimedes on his side assumed:

1.  Equal weights at equal distances are in equilibrium, and equal weights at unequal distances are not in equilibrium but incline towards the weight which is at the greater distance.
2.  If, when weights at certain distances are in equilibrium, something be added to one of the weights, they are not in equilibrium but incline towards that weight to which the addition was made ([11], p. 189).

All the rest of the theory was presented by Euclid and Archimedes as a pure deduction. From the law of the lever, the Hellenistic mathematicians derived the laws of all the machines for lifting weights, or rather of their fundamental components: the pulley, wedge, winch, called simple machines, with the exception of the inclined plane. The solution of the problem posed by this last machine had to wait for the treatises of the Arabs in which a convincing demonstration of the law of the lever was resumed closer to that suggested by the *Mechanica* than by the *De planorum aequlibriis*, and lent itself naturally to the demonstration of the law of the inclined plan. Jordanus de Nemore in the 13th century was the first to exhibit a correct law for the inclined plane. Although not all historians agree, he is credited to have applied the rule: what lifts a weight *np* to a height *h* lifts a weight *p* to a height *nh*, which is a form of the principle of virtual works [12].

The ancient mechanics that in the Middle Ages was called the *Science of weights* remained substantially unchanged until the beginning of the 17th century. It was the discipline developed by mathematicians; therefore, it was of a quantitative nature that studies the equilibrium and displacement of heavy bodies.

## 3. Early Classical Mechanics

With the 17th century, a new perspective opened up, which at the beginning was seen as completely separate from that offered by ancient mechanics. Mathematicians began to be interested in the motion of bodies not only in that occurring with the use of lifting machines but also in the more general case such as the natural motion of falling bodies and the violent motion of an object thrown from a slingshot, whose study since then was carried out by philosophers like Plato, Aristotle in an almost exclusively descriptive way.

The novelty was made possible because mathematicians began to conceive of time as a physical magnitude that could be measured with precision; conversely, one could probably say that the need to treat time as a physical magnitude derived from the need to study the motion of bodies. The main protagonist of this development was Galileo Galilei, who in the 17th century obtained the temporal law of the free fall of a heavy body according to which the space passed downward is in the square ratio of the time.

This law seemed too complex to Galileo to be taken as a principle and he tried to deduce it from a simpler principle. He obtained it by verifying that the law of falling bodies could be derived by assuming the speed of the body increasing linearly with time. Galileo's procedure was quite elaborate—a primordial use of infinitesimal calculus indeed—and it is not completely clear, at least to me, whether it should be considered as a hypothetical deductive approach, for which the law of proportionality of speed and time is tentatively fixed and, if the passed space derived from it is in accordance with the square ratio of the

time, the hypothetically assumed law is valid, or if it followed the analytical method of mathematicians, which derived one or more axioms from a theorem with purely deductive procedures. A modern mathematician would opt for this second solution because the two laws, speed proportional to time and space proportional to the square ratio of time, are obtained from each other with derivation/integration operations.

No less important and perhaps more basic was the formulation of the law of inertia—and the principle of relativity (modern terminology) strictly related to it—based no longer on metaphysical concepts as it had been in the Greek and medieval impetus theory but on thought and real experiments. Galileo's results were motivated by the defense of the heliocentric system and would not have been possible without the publication by Copernicus of the *De revolutionibus orbium coelestium* of 1543 [13].

A very clear exposition of Galileo's principle of inertia is that of the *Dialogo sopra i due massimi sistemi* of 1632, where its experimental—though idealized—nature is shown. In the following exchange, Salviati-Galileo made Simplicius to say that the motion of a body on a horizontal plane, once it has received an initial impetus, would move forever, on the condition that there are not impediments; the motion should occurs without acceleration or delay; it would thus be uniform.

> SIMP. *I cannot tell how to discover any cause of acceleration, or retardation, there being no declivity or acclivity.* [emphasis added]
> SALV. Well: but if there be no cause of retardation, much less ought there to be any cause of rest. How long therefore would you have the moveable to move?
> SIMP. *As long as that superficies, neither inclined nor declined shall last* [emphasis added].
> SALV. Therefore if such a space were interminate, the motion upon the same would likewise have no termination, that is, would be perpetual.
> SIMP. I think so, if so be the moveable be of a matter durable.
> SALV. That hath been already supposed, when it was said, that all external and accidental impediments were removed, and the brittleness of the moveable in this our case, is one of those impediments accidental ([14], p. 173. English translation in [15]).

In the *Discorsi e dimostrazioni matematiche sopra due nuove scienze* of 1638, Galileo added that, in truth, even though all the above mentioned resistances were eliminated, a further resistance would arise to contrast the motion, due to the heaviness of the body. Indeed the horizontal plane is supposed to be represented by a straight line and each point on this line equally distant from the center; this is not the case, for as one starts from the middle of the line and goes toward either end, he departs farther and farther from the center of the earth and is therefore constantly going uphill. Whence it follows that the motion cannot remain uniform through any distance whatever, but must continually diminish ([16], p. 274).

This explanation of Galileo has led many historians to affirm that Galileo did not have a clear idea of the principle of inertia and that his inertia was in fact just a circular inertia ([17,18], p. 352). It seems clear to me that in the affirmation that if resisting forces did not act on a moving body it would continue to move indefinitely, was the purest expression of the principle of inertia, even though it must be said that it is always difficult to establish a correlation between a modern concept and its older formulation. That there are in fact no bodies in nature that can move without resistance, which is another matter. Compared with Newton's formulation, Galileo's differs in one, perhaps non-trivial, way: Galileo did not admit the possibility that there were bodies on which resistant forces did not act, for example weightless bodies; Newton did. Galileo's was a terrestrial view, Newton's a cosmological one. In any case, when Galileo studied the motion of projectiles, he considered a horizontal inertia, without specifying whether he ignored the change in the direction of gravity when moving in a limited space because in this case the direction varies little, or he actually ignored the effect of gravity, thus separating the effect of weight from that of mass.

Cavalieri investigated the motion of projectiles in his *Lo specchio ustorio, overo trattato delle settioni coniche* of 1632 [19]. In this text he had paused to consider the nature of the motions to be combined.

> Moreover I say, that considering that the motion [of the body] is a straight line toward any part, if it had not other motive virtue that would push it towards another direction, [the body] should go in the place indicated by the projector along a straight line, because of the virtue impressed to it was in the straight line, from which direction it is not reasonable that the mobile deviates, as long as there is no other motive virtue that deflects it, and that when between the two terminal points there is not impediment. ([19], p. 154. My translation).

This is an expression of the principle of inertia, closer to Newton than to Galileo. Cavalieri said clearly, in a few lines, that a projectile would move in a straight line when not impeded by something (gravity for instance). The main difference compared to the sentence of Galileo is its explicitness.

Torricelli studied the motion of projectiles in Book II of the *De motu gravium*, written before 1644. For him the composition of motion became just a matter of mathematics, the composition of two motions highly idealized: a downward uniformly accelerated motion and a uniform rectilinear motion ([20], *De motu proiectorum*, p. 156).

The evolution of the principle of inertia due to Cavalieri and Torricelli is an example of the growth of scientific knowledge. A scientist (Galileo) developed a form of the principle of inertia, which he did not elaborate completely. His internal troubles transpired in his work, signaled by some uncertainties in its use. Scientists after him (Cavalieri and Torricelli) who had read his writings were not fully aware of the troubles and applied with no doubt the theory to any possible case (the angled launch of a bullet for instance). The process is similar to what Kuhn refers as gestalt switch [21].

Cavalieri and Torricelli moved ahead of Galileo because their mathematics had moved ahead too. Cavalieri and Torricelli, more inclined to mathematics, did make use of the actual infinite in mathematics; Galileo did not. For them the mathematical physical theory (broad meaning) had its reality, maintaining contact with the external world. For this reason the quasi-rectilinear but still slightly curved trajectories of bullets, moving over the earth sphere, can actually be assumed to be rectilinear and of infinite length, identifying the approximation with reality. However, Cavalieri and Torricelli were more cautious than Newton; for instance, they hardly used the quantifiers *for all* and *for ever*, as Newton did.

The concept of the principle of inertia was substantially revolutionary. It overturned completely the traditional approach to motion according to which it was caused by a force, either internal or external. From now on the force acting on a body is responsible for the change of velocity, and not displacement.

After Galilei, the attention of scientists who studied motion focused on the phenomenon of collision, with important contributions by Descartes and Huygens (with Wren and Wallis). The goal of these scholars, classifiable as mechanistic, was to provide general laws of motion that could explain all phenomena. This project of mechanical philosophy was too ambitious and was essentially aborted. However, new concepts arise from these studies especially thanks to Descartes and Huygens that will prove fundamental at the beginning of the 19th century; those of work and kinetic and potential energy.

Descartes contributed to the development of mechanics mainly as a promoter of a mechanistic philosophy (with Gassendi and Hobbes), simpler than the older philosophy which allowed many people to devote themselves to a quantitative study of nature. He contributed to spread the concept of energy/work in statics (the product of weight and vertical displacement) and in dynamics (the product $m|v|$ of mass and speed), but mainly he was the supporter of a rationalistic view of mechanics, based on evident and clear notions, with no recourse to experience. Huygens played a different but fundamental role, that for the sake of space is not commented upon here. For instance he, with Leibniz (see below), contributed to the development of the ideas of work and energy and formulated

in nuce the concept of conservation of mechanical energy, developed then by Johann and Daniel Bernoulli [22,23].

John Wallis published in 1669–1671 the text *Mechanica sive de motu, tractatus geometricus* [24], where the term mechanics, until then restricted to statics, was also applied to the new science of motion. Aside from a greater wealth of content, mechanics took on a different function. It was no longer a discipline destined to solve a particular problem, of mainly technological interest, but became a new philosophy of nature with which the mathematical philosophers (that is mathematicians) and not the generalist philosophers intended to explain nature.

## 4. Classical Mechanics

The transformation of mechanics into a new philosophy of nature was particularly evident with Newton's *Philosophiae naturalis principia mathematica* of 1687 [25], which took up the work of Archimedes and Galileo by unifying statics and dynamics with few very simple axioms: the three famous laws of motion, considered certain and inductively derivable from experience. The approach was exactly that of the mixed mathematics of the ancient Greece, nothing substantially new after two thousand years. However, the mathematics was more refined, using modern categories it can be said that the *Principia* was a treatise of differential geometry.

Below are the first two laws of motion as reported in the 1726 edition.

Law I.
*Everybody perseveres in its state of being at rest or of moving uniformly straight forward, except insofar as it is compelled to change its state by forces impressed.*
Law II.
*A change in motion is proportional to the motive force impressed and takes place along the straight line in which that force is impressed.* ([26], p. 13. English translation in [27]).

Notice that Newton's second law is quite different from what is known today under this name. The modern relation, force equals mass by acceleration, is due mainly to Euler (see below).

Modern philosophers of science have clearly shown that Newton's laws/axioms cannot be derived from experimental data by induction, not least because many of them do not believe that induction exists. However, Newton and his followers were convinced of the empirical derivation of the laws and that all natural phenomena could be derived from them with certainty.

The comment of Ernest Nagel on the nature of Law I is worth reporting. According to him, it is clear that the first law of motion, taken in isolation, is seriously incomplete as a statement intended to have empirical content. To say that a body will persevere in its state of rest or uniform motion in a straight line, unless compelled to change its state by forces impressed on it, is to say nothing definite, if nothing further is stated as to:

1.  What is the spatial frame of reference;
2.  What is the way to measure time;
3.  What are the marks by which the presence or absence of impressed forces is determined.

Assumed that the first point will in some way be solved, continued Nagel, what is the criterion for asserting that a body is under the action of no forces? What defines the perseverance of the body in its state of rest or uniform motion in a straight line? However, in such a case, the law is a concealed definition, a convention which specifies the conditions under which one will say that there are no impressed forces acting on a body. Moreover, in addition to assuming a definite spatial frame of reference, in its usual formulation the law takes for granted a definite system of chronometry. If, then, some method not involving the explicit or tacit use of the law were available for identifying the absence of forces, the law could be construed as an implicit definition either of uniform motion or equal time intervals ([3], pp. 175–178).

The exposition of the three laws was followed by some examples that made them plausible even though there was no reference to their empirical nature. Only in the scholium did Newton say that his laws were accepted by mathematicians and confirmed by numerous experimental results. They are attributed to Galileo

Scholium The principles I have set forth are accepted by mathematicians and confirmed by experiments of many kinds. By means of the first two laws and the first two corollaries [the law of composition of forces] Galileo found that the descent of heavy bodies is in the squared ratio of the time and that the motion of projectiles occurs in a parabola, as experiment confirms, except insofar as these motions are somewhat retarded by the resistance of the air ([26], p. 21. English translation in [27]).

This attribution is generally considered too generous on Newton's part. In fact, Galileo had not possessed the concept of impressed force and, if he had read the *Principia*, most likely he would not have understood them. However, it should be noted that, in the 18th century, mathematicians who took up the theory, such as Varignon and Euler for instance, referred to Newton's laws as the laws of motion of Galileo.

Newton considered only the third law as original of him:

Law III.
*To any action there is always an opposite and equal reaction; in other words, the actions of two bodies upon each other are always equal and always opposite in direction* ([26], p. 14. English translation in [27]).

In addition to its explanations, thought experiments were reported; the simplest of them is described below.

I demonstrate the third law of motion for attractions briefly as follows. Suppose that between any two bodies A and B that attract each other any obstacle is interposed so as to impede their coming together. If one body A is more attracted toward the other body B than that other body B is attracted toward the first body A, then the obstacle will be more strongly pressed by body A than by body B and accordingly will not remain in equilibrium. The stronger pressure will prevail and will make the system of the two bodies and the obstacle move straight forward in the direction from A toward B and, in empty space, go on indefinitely with a motion that is always accelerated, which is absurd and contrary to the first law of motion ([26], p. 25. English translation in [27]).

However, the results of a real devised experiment were also reported.

I have tested this with a lodestone and iron. If these are placed in separate vessels that touch each other and float side by side in still water, neither one will drive the other forward, but because of the equality of the attraction in both directions they will sustain their mutual endeavors toward each other, and at last, having attained equilibrium, they will be at rest ([26], p. 25. English translation in [27]).

With his laws, Newton was able to prove old results and to find new ones. The proof of the ellipticity of the orbits of the planets about the sun, as predicted by Kepler, and the formulation of the law of universal gravitations are very interesting.

## 5. Post Newtonian Mechanics

Newton's mechanics proved to be incomplete as it was substantially limited to the motion of free mass points. This was particularly clear to the handful of brilliant mathematicians of the 18th century, of which the most known are Leibniz, Johann Bernoulli, Varignon, Hermann, Euler, d'Alembert, Lagrange that were able to fill the gap [28]. In the following, for the sake of space, a short hint to the contributions of Leibniz, Euler and Lagrange only is given.

Leibniz played a role similar to that of Descartes—both were great philosophers and mathematicians. His main contribution to mechanics was a critical discussion about Newton's concepts: force, time and space, the introduction of the term *dynamics* to indicate that

part of mechanics devoted to the study of the motion of bodies, and, mainly, the development of a form of infinitesimal calculus more efficient in mechanics than that proposed by Newton. It is worthy signaling however that Leibniz's name, though his role is well recognized by historians, is hardly found in modern textbooks of mechanics.

Euler is usually considered as Newton's most important heir. His main merit is of having transformed mechanics from a geometric discipline into an analytical one, considered as a purely rational discipline. The incipit of his early treatise *Mechanica sive motus scientia analytice exposita* of 1636 are quite famous:

Thus, I always have the same trouble, when I might chance to glance through Newton's *Principia* or Hermann's *Phoronomia*, that comes about in using these [synthetic methods], that whenever the solutions of problems seem to be sufficiently well understood by me, that yet by making only a small change, I might not be able to solve the new problem using this method. Thus I have endeavored a long time now, to use the old synthetic method to elicit the same propositions that are more readily handled by my own analytical method, and so by working with this latter method I have gained a perceptible increase in my understanding. Then in like manner also, everything regarding the writings about this science that I have pursued, is scattered everywhere, whereas I have set out my own method in a plain and well-ordered manner, and with everything arranged in a suitable order. Being engaged in this business, not only have I fallen upon many questions not to be found in previous tracts, to which I have been happy to provide solutions, but also I have increased our knowledge of the science by providing it with many unusual methods, by which it must be admitted that *both mechanics and analysis are evidently augmented more than just a little* ([29], Preface. English translation by Bruce I).

Notice that the title of Euler's treatise paraphrased that of Wallis, by changing *geometric* with *analytic*.

Apart from the algebraization of mechanics Euler improved much the 'primeval' Newton's development. For instance he contributed to give the second law of motion its modern form [30,31]. Moreover he applied the laws of motion to the case of extended and constrained bodies, fluid included. Euler was an extremely prolific and methodical mathematician. With him, mechanics became the discipline we know. And even today it makes sense to read some of his writings on mechanics, finding interesting topics for current research.

After Euler mechanics had become an algebraic theory; in some parts however geometry was still present, for example in the use of vectors. A concept, that of vector, today possibly introduced without any reference to classical geometry but that in the 18th century could not. Lagrange with his *Méchanique analitique* of 1788 [32] (Since the second edition of 1811 the title of Lagrange's treatise changed to *Mécanique analytique*.) undertook an important and big step toward a full algebraization of mechanics. His treatise of 1788 added little new to the development of mechanics and collected most results obtained by Lagrange himself since the 1760s. However, it was completely new because of its way of presentation and its logical–epistemological conception. Mechanics then became analytical.

The locution analytical mechanics had/has different meanings. Today its more diffuse acceptation is that of a (mathematical) theory based on algebra and Calculus, whose axioms are presented with scalar relations; this definition substantially covers *Méchanique analitique* use. According to Truesdell, analytic mechanics is limited to discrete systems (excluding thus continua, solid and fluid), and this limitation of the term is quite diffuse also [33]. Vectors could be still present, but they were introduced only through their components and mainly there could be many different local frames in the same mechanical assembly. According to the meaning assumed beforehand, that of Euler is not analytical mechanics, even though he himself referred to it as such.

In the preface of his masterpiece, Lagrange stated that his mechanical theory had become a branch of analysis. The axioms of this branch were represented by general

formulas, the principle of virtual work and the principle of d'Alembert. They were enough to solve any practical problem, both in statics and dynamics. The axioms were considered to be given, and no serious interest was addressed to their derivation or proof. Accordingly the *Méchanique analitique* should be considered one of the first texts of modern mathematical physics.

In the following a large quotation of the preface of Lagrange's treatise is reported, even though well known in the literature, because of its relevance.

> I propose to condense the theory of this science and the method of solving the related problems to general formulas whose simple application produces all the necessary equations for the solution of each problem. I hope that my presentation achieves this purpose and leaves nothing lacking. In addition, this work will have another use. The various axioms presently available will be assembled and presented from a single point of view in order to facilitate the solution of the problems of mechanics. Moreover, it will also show their interdependence and mutual dependence and will permit the evaluation of their validity and scope. I have divided this work into two parts: statics or the theory of equilibrium, and dynamics or the theory of motion. In each part, I treat solid bodies and fluids separately. *No figures will be found in this work* [emphasis added]. The methods I present require neither constructions nor geometrical or mechanical arguments, but solely algebraic operations subject to a regular and uniform procedure. *Those who appreciate mathematical analysis will see with pleasure mechanics becoming a new branch of it* [emphasis added] and hence, will recognize that I have enlarged its domain ([32], Preface. English translation in [34]).

Lagrange's and Euler approaches were different in two main aspects. First, Euler's mechanics had a substantially axiomatic structure, in which concepts were explained before being used. Lagrange's mechanics did not have such an axiomatic structure. It generally used concepts and axioms already accepted by experts of mechanics. Moreover the general formulas Lagrange talked about were not true formulas but rather rules that needed interpretation. Second. Lagrange's mechanics made no reference to natural philosophy to introduce its assumptions, differently from Euler's. 'Axioms' were justified with mathematical reasoning. Where it would be necessary to exit from this ambit, Lagrange was elusory. Instead of considerations drawn from natural philosophy or metaphysics, he referred to a historical account [35]. History gave a justification of the theory and replaced the justification based on natural philosophy. Lagrange thought that mechanics were nothing but the results gained in its history, by Archimedes *in primis*. In his history there was no room for the philosophy of nature.

## 6. The Crisis of Classical Mechanics

In the 19th century, mechanics was a mature discipline, or at least it was considered as such by its experts. There were no substantial innovations in the understanding of the relevant phenomena. One of the main 'discoveries' was the Coriolis effect, that is the fact that in a reference system in rotation with respect to the fixed stars, apparent forces orthogonal to the motion are noted. The effort of mathematicians and physicists was concentrated on perfecting the formal aspects of the theory and in the criticism of its axioms.

At the turn of the century, after the publication of Lagrange's *Méchanique anlitique*, there was a heated debate on the logical status of the principle of virtual works, which was at the basis of Lagrange's analytical mechanics. The protagonists of this discussion were the French mathematicians and physicists, including Fourier, Ampère, Coriolis, Laplace and Poinsot ([12], pp. 317–351).The discussion showed how difficult it was to 'prove' in a rigorous way, that is according to the stringent standard of rigor of the 19th century, axioms that appeared nearly self-evident to the common people, as the composition of forces, the law of lever, the principle of virtual works. For this purpose the words by Lazare Carnot at the very end of his *Essai sur les machines en général* of 1782/86 are illuminating.

> Sciences are as a beautiful river whose course is easy to follow, when it has acquired a certain regularity; but if one wants to sail to the source one cannot find it anywhere, because it is far and near; it is diffuse somehow in the whole earth surface. The same if one wants to sail to the origin of science, one finds nothing but darkness and vague ideas, vicious circles; and one loses himself in the primitive ideas ([36], p. 107).

In the second half of the century, the analysis of the foundations concerned the whole mechanics; in particular, the Newtonian force was the subject of criticism. The protagonists were mathematicians, physicists and philosophers, such as: Saint-Venant, Kirchhoff, Hertz, Helmholtz, Reech, Mach, Poincaré, Duhem ([37], pp. 417–443).Particularly effective were the contributions of the latter three scholars. Mach in his *Die Mechanik in ihrer Entwicklung historisch-kritisch dargestellt* of 1883 scrutinized all the ideas of Newtonian mechanics, starting from the ideas of absolute space, force and mass [38]. Poincaré in the *Science et hypotèses* of 1902, commented upon the epistemology of the axioms of mechanics. According to him, mechanics is neither fully empiric nor fully a priory. For instance let us consider the principle of inertia:

> *The Principle of Inertia—A body under the action of no force can only move uniformly in a straight line*. Is this a truth imposed on the mind à priori? If this be so, how is it that the Greeks ignored it? How could they have believed that motion ceases with the cause of motion? or, again, that every body, if there is nothing to prevent it, will move in a circle, the noblest of all forms of motion. [. . . ] Is, then, the principle of inertia, which is not an à priori truth, an experimental fact? Have there ever been experiments on bodies acted on by no forces? and, if so, how did we know that no forces were acting? ([39], pp. 112–113).

Beside the attempts to clarify the axioms, there was also the problem of the nature of mechanics in itself. Until then mechanics had been considered as the mother of the other disciplines; even electricity could be framed into a mechanical context. However, at the end of the 19th century it was clear that this was not a tenable position. Thermodynamics dealt with concepts hardly reducible to mechanics, and a similar consideration held good for electromagnetism.

Duhem championed 'energetics', holding generalized thermodynamics as foundational for physical theories; that is, thinking that all of chemistry and physics, including mechanics, electricity, and magnetism, should be derivable from first axioms of thermodynamics. His *Traité d'énergétique ou de thermodynamique générale* of 1911 [40] represented one of the first attempts to establish a physical theory on modern axiomatic basis. The author believed that the axioms of a physical theory did not require any justification apart from the assessment of their internal consistency. He considered, however, that the axioms could not be chosen by chance, but that they should benefit from formulations of past similar axioms; in this way the history of science becomes an integral part of science ([40], pp. 183–246).

The attempt by Duhem to save classical mechanics was frustrated by the emergence of serious paradoxes in the electromagnetic theory, still framed into classical mechanics for some aspects, evidenced at the end of the 19th century. They pushed to consider classical mechanics as a simple approximation of a true mechanics, the relativistic one. From then on, physicists ceased to be interested in classical mechanics, considering it scarcely interesting, or worse a dead discipline. However, this is another story.

Together with the intense work on foundation there was also an effort to clean up the theory. Hamilton and Jacobi moved in this direction, giving analytical mechanics its modern structure, based on a functional, well-known as the Hamiltonian, which allowed the treatment of statics and dynamics, rigid and continuous deformable solid bodies and fluids in a unitary way. With them the formal aspects, the elegance, the clarification of the spheres of validity of the theorems had reached the apex [41,42]. The rearrangement of mechanics also made the solutions of some problems easier and also suggested the solution of others; it is so restrictive to speak of formal perfecting only. Mechanics was now ready to be framed into the broad field of modern mathematical physics.

The expression mathematical physics had substantially two different meanings. On the one hand, it simply indicated modern physics, strongly rooted in mathematics; in this sense, the great mathematical physicists were Galileo, Kepler, Newton, Euler, etc. On the other hand, it meant that branch of science born in the 19th century which afforded some problems regulated by partial differential equations, e.g., the propagation of heat, the theory of potential, the theory of elasticity; in this sense, great mathematical physicists were Fourier, Lamé, Gauss, Piola, Beltrami, etc. Today the expression indicates an academic discipline cultivated by mathematicians which has as its basis some physical axioms whose merit was not questioned.

Examining epistemological aspects of mathematical physics gives the opportunity to stipulate an appropriate conventional meaning. As a first step, it is useful to explain the essence of physical theories. They are made up of three parts:

1.  An abstract calculus which comprehends theoretical terms, definitions, axioms and inference rules.
2.  A conceptual model which more or less gives a representation of the world.
3.  Some correspondence rules connecting theoretical terms and theorems of the theory with the external world [22].

When the correspondence rules are missing, the physical theory turns to a mathematical physical theory as its axioms speaks about the physical world and not only of pure mathematical entities. The boundary between pure mathematics and mathematical physics is not exactly specified and various positions could be assumed. Clifford Ambrose Truesdell, for instance, does not see any difference and claims mathematical physics to be simply a branch of pure mathematics. He actually refers to his field of competence, rational mechanics, but his considerations apply to any mathematical physical theory:

> Is rational mechanics part of applied mathematics? Most certainly not. While in some cases known mathematical techniques can be used to solve new problems in rational mechanics, in other cases new mathematics must be invented. It would be misleading to claim that each achievement in rational mechanics has brought new light in mathematics as a whole as to claim the opposite, that rational mechanics is a mere reflection from known parts of pure mathematics ([43], p. 337).

Other people sustain that a theory can be called mathematical only if it is addressed to purely mathematical objects.

Quite interesting in the development of modern mathematical physics was the contribution by Carl Neumann [44]. He recognized that the theorems of mathematical physics should be confirmed by experimental data ([45], p. 130), but as a mathematician (or mathematical physicists) he found that he should focus his attention on the mathematical description of axioms and on the improvement of mathematical tools ([45], p. 127). In his time there was a substantially Aristotelian–Euclidean vision of mathematics, according to which a mathematical theory should be based on indubitable axioms so that the resulting theorems could not be disputable. According to Neumann, a mathematical physical theory was different from a pure mathematical theory because in it some 'axioms' might be hypothetical, and consequently its theorems could not necessarily be true. This possibility made mathematical physical theories very interesting for a mathematician because of their greater potential of invention (hypothetical deductive theories). These considerations by Neumann, shared by Mathieu ([46], pp. 110–111), contributed to the development of the modern concept of a mathematical theory based on arbitrary premises.

The approach by Neumann and Mathieu, not very different indeed from that of Duhem, paralleled a similar process emerging in mathematics, that will continue in the 20th century. After moving away from the Euclidean rigor for a long time, mathematics had returned to it. Therefore, demonstrations of many properties that were previously considered evident were required; indeed, this was in many cases the only way to discover the limits of their validity. The concepts of function, continuity, limit, infinity had revealed the need for a more precise determination; the negative number and the irrational number,

which have long since become part of mathematics, have had to be subjected to a more precise examination of their justification. Thus, there was the tendency to give rigorous proofs everywhere ([47], p. 1).

In geometry, the attempts to prove the postulate of parallels (fifth Euclid's postulate) from the other postulates proved to the contrary its independence from them. This suggested to Lobachevsky and Bolyai the idea to replace it with a different assertion on the parallels: they could meet. This gave rise to the so called non Euclidean geometry. This form of geometry furnished results in contrast with many points with the Euclidean geometry, even though the forms of reasoning were similar and suggested the possibility of creating a mathematical theory in which the assumption of the truth of axioms was not required [48,49].

The new theories of mathematics or mathematical physics were hypothetical deductive. It is true that since the ancient Greeks there had been theories based on hypotheses and at large to be classified as hypothetical deductive, but there the hypothesis could not be fully arbitrary; they were suggested in some way by observations or deductions and were considered to assume a value of truth if the theory gave true results. Modern mathematical physics has released this requirement.

**Funding:** This research received no external funding.

**Conflicts of Interest:** The author declares no conflict of interest.

**Entry Link on the Encyclopedia Platform:** https://encyclopedia.pub/12056.

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
