# Peer review of "The Foundation of Classical Mechanics"

_encyclopedia, doi:10.3390/encyclopedia1020040_

Round 1

Reviewer 1 Report

In this article, the author introduces the foundation of classical mechanics from historical and philosophical aspects. The difference between the complete axiomatized formulation and the incomplete axiomatized formulation of classical mechanics is discussed. The contents range from ancient mechanics to post Newtonian mechanics. I think the article is well written, it is inspiring not only for the general reader and also for researchers of physics. I can only find a minor typo as the following:

At the end of paragraph 4 on page 2: a comma is missing. "masses, forces times, ..." should be "masses, forces, times, ...". 

Author Response

Thank you very much  for your reading and comments.

Reviewer 2 Report

It is difficult to find any objective history of mankind's achievements in any field of activity since each presentation of such a story is necessarily marked by the subjective point of view of the author of a given narration. A very blunt opinion in this regard is attributed to Napoleon Bonaparte, who was to state that "history is a set of lies that people have agreed upon". According to Piero Villaggio [A] history of science can be associated with "the selective accumulation of evidence from centuries of scientific endeavor". That is why it is so important who and how makes this selection, and how deeply his studies of literature sources go. In the case of the reviewed work, we are dealing with outstanding work, prepared by a very competent Author, who supports his own opinions with quotations from many important works recognized as milestones in the development of mechanics. The above statement does not mean, however, that presented by prof. Capecchi's look at the "foundation of classical mechanics" will not be the subject of lively critical discussion among specialists. For that to happen, however, the work should be published, which I highly recommend.

However, I would like to point out that the current version of the manuscript requires a linguistic correction, among others in terms of standardizing the English version used (US / GB, e.g. "signaled" / "signalled"), or minor typos: "recurse" / "recourse", "brittlenesse" / "brittleness", "self evident"/"self-evident", "every day"/"everyday","gave raise"/"gave rise","subbject"/"subject", "mechanic"/ "mechanics","algebrization"/ "algebraization","had became" / "became" (or "had become"), etc.

Referring to the anticipated discussion on this article, let me express my very subjective surprise at the lack of any reference in the text to some interesting publications dealing, mostly, with the structural analysis, but with clear references to the history of mechanics [B-G].

Additional readings:

[A] Villaggio, P. (2001). Distorsions in the history of mechanics. Meccanica, 36(5), 589-592. doi:10.1023/A:1015662624992

[B] Benvenuto, E. (1991) An Introduction to the History of Structural Mechanics. Part I: Statics and Resistance of Solids, Springer-Verlag, New York, doi:10.1007/978-1-4612-2982-7

[C] Becchi, A., Corradi, M., Foce, F., Pedemonte, O. (eds.) (2003) Essays on the History of Mechanics. In memory of Clifford Ambrose Truesdell and Edoardo Benvenuto, Birkhäuser Verlag, Berlin, ISBN: 3764314761

[D] Maltese G. (2006) On the Changing Fortune of the Newtonian Tradition in Mechanics. In: Williams K. (eds) Two Cultures. Birkhäuser Basel. https://doi.org/10.1007/3-7643-7540-X_8 

[E] Kurrer K.-E. (2012) The History of the Theory of Structures: From Arch Analysis to Computational Mechanics, John Wiley & Sons, ISBN: 9783433601341 

[F] Allen, D. H. (2014). How mechanics shaped the modern world, Springer, ISBN: 9783319017013 

[G] Dixit U.S., Hazarika M., Davim J.P. (2017) History of Mechanics. In: A Brief History of Mechanical Engineering. Materials Forming, Machining and Tribology. Springer, Cham. https://doi.org/10.1007/978-3-319-42916-8_3 

Author Response

Thank you very much  for your reading and suggestions. You are right, there are many citations missed. This is mainly due to reason of space.

The references you proposed are well known to me, apart the nice book by Allen. I reflected however on your suggestion and decide to cite Maltese (a scholar very appreciated by me) and Allen.

Reviewer 3 Report

The reviewer would like to thank to authors for this nicely written article. before publication please consider the folowing points: 

--In the definition : typo in "motion in bodies subbject to" typo should be corrected. 

"--The comment of Ernest Nagel on the nature of Law I are " are should be is. 

--"and the brittlenesse of the moveable "

Author Response

Thank you very much  for your reading and comments.  Modifications have been done.

Reviewer 4 Report

The paper is very clearly written and includes the most important turning points of classical mechanics.

Remarks:
- the achievement of Nicolaus Copernicus should be added

Author Response

Thank you very much  for your reading and comments. Regarding Copernicus I must confess that was for a lack of concentration I have forgotten him. This is because Copernicus gave no direct contribution to foundation of mechanics. But his heliocentric system influenced much scholars of mechanics, Galileo in primis. I added some comments and a citation.

Reviewer 5 Report

The English should be double checked “character of the phenomena it studies, it had to” and other formulations.

This sentence “Let us follow, for instance, a projectile” seems incorrect formulation in the paragraph context as this normally happen now and then you speak about mechanics from past.

Incorrect statement “Despite the unusually distinguished and successful role classical mechanics has played

in the history of modern science”

I suggest to replace the introduction with brief history, and indeed create a shortly introductory part as introduction otherwise not clear how this work is structured and which is the benefit of researchers/readers.

Please define all the acronyms before their first appearance in text.

Please provide some citation for “The approach by Neumann and Mathieu”.. “suggested

to Lobachevsky and Bolyai”

some perspective will make this work much more attractive.

I have looked briefly over similarities reports and I suggest to authors to reduce it. Even many of these sentences which contain similarities were cited it will be better to rephrase them at list one which are not old authors lema.

Author Response

Thanks for your suggestions and criticisms, that I greatly appreciated even though I do not agree completely on them.

Regarding the defects of my English, I think you are right; I am not a native speaker. I revised however the paper, that hopefully could be a little better.

You suggest to open my paper with a historical introduction. I agree this is a possibility, but I consider it more convenient to start with a general presentation of the subject as seen presently. Of course this approach may lend to a whig history, but I hope this is not the case for the present paper.

About the possibility to rephrase and summarizing texts by eminent scientists or historians (I am not sure to have understood what you mean), I concede it is a good approach in many situations, especially in a short paper as mine. My personal opinion is however that it is always better a verbatim quotation to avoid possible ambiguities, and it has become a habit of mine.

Of course, some more space should be devoted to Neumann, Mathieu, Bolyai and Lobachevski; but it is true for many other scientists. I added new references, however.

Round 2

Reviewer 2 Report

I am pleased to note that in the revised version some minor linguistic shortcomings highlighted in my previous review have been corrected and, furthermore, the list of publications cited has been extended as I suggested. I have no choice but to recommend the publication of this work in the Encyclopedia.

Reviewer 3 Report

Thank you for your modifications. Its a well written article.

Reviewer 4 Report

Thank you.

Reviewer 5 Report

-